# Before or Concomitant Drinking Greenleaf Juice with Rice Reduces Postprandial Blood Glucose Levels in Healthy Young Women

**DOI:** 10.3390/nu16193226

**Published:** 2024-09-24

**Authors:** Nobuko Sera, Fuka Taguchi, Isaki Hanamura, Ryoko Hongo

**Affiliations:** Department of Nutrition Science, Faculty of Nursing and Nutrition, University of Nagasaki, Nagasaki 852-8131, Japan; bn219048@sun.ac.jp (F.T.); hanamura02@sun.ac.jp (I.H.); hongo@sun.ac.jp (R.H.)

**Keywords:** greenleaf juice, postprandial blood glucose, intake-timing

## Abstract

The purpose of this study was to examine how green leaf juice drinking affect the postprandial blood glucose. Postprandial hyperglycemia causes vascular endothelial damage and chronic inflammation, promoting atherosclerosis, regardless of the presence of diabetes. Some ingredients in greenleaf juice have been reported to suppress blood glucose levels; however, the effect of greenleaf juice on reducing blood glucose levels in healthy individuals is unclear. We observed changes in postprandial blood glucose levels in 13 healthy young women who drank greenleaf juice before or concomitantly with rice. Compared to water, greenleaf juice consumption reduced blood glucose levels at 90 and 120 min after rice consumption, with no difference regardless of the time of greenleaf juice consumption. Greenleaf juice may be one of the most convenient and cost-effective methods for reducing postprandial blood glucose in healthy people.

## 1. Introduction

Postprandial hyperglycemia is defined as a blood glucose level of 140 mg/dL or more two hours after meals (International Diabetes Federation; Guidelines 2007), causes vascular endothelial disorders and chronic inflammation, and promotes arteriosclerosis to develop cerebral infarction and ischemic heart disease [1]. Oscillating glucose is deleterious to endothelial function and causes oxidative stress in individuals with and without diabetes [2]. Therefore, reducing postprandial blood glucose levels not only improves glycemic control in diabetics and suppresses the onset and progression of various diabetic complications, such as neuropathy, retinopathy, and nephropathy, but also reduces the risk of macrovascular complications in patients with no diabetes.

Polyphenols, flavonoids, phenolic acids, anthocyanins, and proteins present in plant-based foods have been identified as the major components responsible for antioxidant and other biological activities. In rats, it has been reported that greenleaf juice delays carbohydrate breakdown in the intestinal tract by inhibiting α-amylase and α-glucosidase activities in the pancreas and intestines, respectively, and suppresses postprandial blood glucose levels [3]. Greenleaf juice contains vitamins, minerals, and dietary fiber, mainly made from green leaves, such as young barley leaves (*Hordeum vulgare* L.) and kale (*Brassica oleracea* var. *acephala*). Young barley leaf, the main ingredient, has been reported to reduce postprandial blood glucose levels in rats without diabetes mellitus [4]. Additionally, kale has been reported to promote adiponectin secretion in adipocytes and increase insulin sensitivity [5]. The water-soluble dietary fiber contained in greenleaf juice has a gelling effect, delays gastric emptying and transit of food in the intestinal tract, and inhibits carbohydrate absorption [6]. In a previous study in humans, vegetable salads, vegetable juice, or tomato juice intake 30 min before meals significantly reduced postprandial blood glucose levels [7,8,9]. However, there are few reports on the effects of greenleaf juice consumption on postprandial blood glucose levels in humans.

This study examined the effects of greenleaf juice on postprandial blood glucose levels by examining two patterns of intake timing: simultaneously with rice and 30 min before.

## 2. Materials and Methods

### 2.1. Study Participants

Volunteers were recruited at the University of Nagasaki, Japan. The participants were 13 healthy women aged 18–23 years. The exclusion criteria were (1) individuals with type 1 or 2 diabetes and impaired glucose tolerance on medical examinations and (2) body mass index (BMI) > 25 kg/m^2^. This study complied with the ethical guidelines of the Declaration of Helsinki. The study was approved by the Ethics Committee of the University of Nagasaki (date of approval: 31 March 2022, approval no. 501), and informed consent was obtained from all participants.

### 2.2. Test Foods

This study used greenleaf juice per 200 mL bottle (ITO EN Co., Ltd., Tokyo, Japan). The main components are barley grass and kale, which do not contain sugar and fat. The nutrient information for the greenleaf juice is shown in Table 1. For the meal, commercially available cooked packaged rice “Sato no Gohan” (Sato Food Industry Co., Ltd., Niigata, Japan) was used. The test meal in each study was a combination of (1) a concomitant intake of 200 mL of water and 150 g of rice, (2) a concomitant intake of 200 mL of greenleaf juice and 150 g of rice, (3) drinking 200 mL of water 30 min before consuming 150 g of rice, and (4) drinking 200 mL of greenleaf juice before consuming 150 g of rice. The amount of nutrients contained in the test meal was calculated by quoting the information on the package (Table 2).

### 2.3. Study Methods

The participants were prohibited from eating and drinking anything but water for more than 10 h from 9 P.M. on the night before the experiments until the start of the experiments. In test meals (1) and (2), each meal was consumed from 5 to 10 min simultaneously, and the blood glucose level before and after the test meal was measured at seven time points (0, 15, 30, 45, 60, 90 and120 min after ingestion). In test meals (3) and (4), rice was ingested (0 min) 30 min after drinking water or greenleaf juice (−30 min), and the blood glucose level was measured at eight time points (−30, 0, 15, 30, 45, 60, 90 and 120 min after ingestion). Each trial was conducted at least two days apart. All participants were required to wear a continuous blood glucose meter (Freestyle Libre; Abbott Japan, Tokyo, Japan) for the continuous measurements of tissue glucose levels.

### 2.4. Data Analysis

Changes in blood glucose levels (Δglucose) were calculated by subtracting the glucose value measured before intake of the test meals from the values measured at the various time points. The maximum glucose value (ΔGmax) is set to the maximum change in Δglucose, and the incremental area under the curve (IAUC) was calculated by a trapezoidal formula from the transition of Δglucose. Statistical analyses were performed using SPSS software (Statistics version 28, Tokyo, Japan). Glucose levels at the start of the test (fasting blood glucose), ΔGmax, IAUC, and Δglucose, the four groups, were compared using a one-way analysis of variance using Turkey–Kramer test. In addition, Δglucose in two groups with the same study design, (1), (2) and (3), (4), were compared using paired *t*-tests. In the Δ-glucose analysis between the two groups, Bonferroni correction was performed on the obtained *p*-values to examine the multiplicity of tests. *p* < 0.05 was considered statistically significant. The results were provided as mean ± standard error (SE). 

## 3. Results

Thirteen healthy women (age: 20.5 ± 1.5 years, body weight: 53.6 ± 5.6 kg, BMI: 22.2 ± 2.2 kg/m^2^) enrolled and completed the study. The effect of simultaneous intake of greenleaf juice and rice on reducing postprandial blood glucose levels was compared between test diets (1) and (2). Δglucose levels were significantly lower after 90 min if greenleaf juice was simultaneously consumed with rice than water (13.4 ± 4.2 vs. 31.1 ± 5.4, *p* = 0.030) (Table 3, Figure 1). Fasting blood glucose (FPG) and ΔGmax, IAUC did not differ (Table 4). The effect of greenleaf juice on reducing postprandial blood glucose levels before or concomitant with rice intake was compared between test diets (3) and (4). Compared to the group that drank water 30 min before rice consumption (4), the greenleaf juice group (3) had significantly lower Δglucose after 120 min (6.1 ± 2.7 vs. 22.9 ± 4.0, *p* = 0.028) (Table 5, Figure 2). There was no difference in FPG, ΔGmax, and IAUC (Table 6). There was no significant difference in each postprandial Δglucose, ΔGmax, or IAUC for concurrent intake of green leaf juice with rice (2) and 30 min before (4) 

## 4. Discussion

Postprandial hyperglycemia is a risk factor for cardiovascular disease [1], and prospective cohort studies of European and Asian men and women have reported that blood glucose levels 120 min after glucose loading are more indicative of cardiovascular disease and mortality than fasting blood glucose [10]. In patients with normal glucose tolerance, the blood glucose level at 120 min after meals is approximately <140 mg/dL [11], and the IDF Guidelines 2007 also define postprandial hyperglycemia as 140 mg/dL or more at 120 min after meals. In the diagnostic criteria for diabetes in Japan, the 120-min value of the 75 g glucose tolerance test is defined as <140 mg/dL. Currently, there are approximately 10 million individuals who have impaired glucose tolerance in Japan, and acarbose, an α-glucosidase inhibitor, is covered by insurance as a treatment for individuals with impaired glucose tolerance. Acarbose reduces the incidence of type 2 diabetes [12]; however, its effectiveness in reducing the risk of developing cardiovascular diseases is inconsistent [12,13]. In Japan, lifestyle correction is prioritized for individuals with impaired glucose tolerance, of which only a few consume acarbose. Therefore, in patients with impaired glucose tolerance, besides a diet that focuses on proper energy intake and nutrient balance, some individuals practice methods to suppress postprandial hyperglycemia, such as the ingestion of foods containing protein (meat and fish) or vegetables, before carbohydrates (rice, bread, and pasta) ingestion [7] and consume functional foods that are said to suppress postprandial hyperglycemia, such as Jerusalem artichoke [14], tea catechin [15], and mulberry leaf [16]. Although there are not many reports of postprandial blood glucose in healthy subjects, 9.7% of healthy women (mean age 50 years) had hyperglycemia of 140 mg/dL for more than 6 h a day [17], suggesting that there are more than a certain number of people who have cardiovascular risk without being aware of it.

Greenleaf juice, commercially available in Japan, contains various vegetable nutrients, such as dietary fiber, vitamins, and minerals, and is a healthy food that supplements the nutrition of vegetables. Moreover, among the raw materials and components of greenleaf juice, dietary fiber [6], kale [5,18], and barley grass [4] have been reported to suppress postprandial blood glucose levels. However, there have been few reports on the effects of greenleaf juice intake on postprandial blood glucose levels in humans. In this study, postprandial blood glucose was significantly lower at 90 min after a meal if concurrently consuming greenleaf juice with rice and 120 min after rice ingestion when consuming green juice 30 min before. Conversely, there was no difference in postprandial blood glucose, ΔGmax, and IAUC owing to differences in the intake time of greenleaf juice between before or concomitantly with rice. There are several factors that may have been involved in reducing postprandial blood glucose levels by greenleaf juice. The mechanism of the effect of water-soluble dietary fiber on suppressing blood glucose levels decreases the rate of gastric emptying [19,20], which is observed if ≥5 g of indigestible dextrin is ingested simultaneously [21]. It has been reported that the intake of 7.5 g of water-soluble dietary fiber suppresses postprandial blood glucose levels in patients with diabetes, but this has not been observed in healthy participants [19]. Here, the amount of dietary fiber contained in the greenleaf juice was 0.4–1.4 g, and since the participants were healthy, it is unlikely that dietary fiber was involved in postprandial blood glucose elevation. As mentioned in the Introduction, in animal experiments using greenleaf (amaranthus, rumex, or sowa) juice, the inhibition of α-glucosidase activity in the intestines or α-amylase activity in the pancreas was revealed, and the induced blood glucose of 30–120 min after meals was generally low, and the IAUC was significantly lower [3]. The effect of reducing postprandial blood glucose levels in healthy subjects may be influenced by the delay in the carbohydrate breakdown by antioxidants contained in barley grass and kale and the increased insulin sensitivity of kale. Furthermore, the effect does not differ between before or concomitantly with rice, and it appears that it acts relatively long after greenleaf juice consumption. Regarding the effect of suppressing postprandial blood glucose levels by consumption of vegetable juice 30 min before meals, it has been reported that the sugar contained in vegetable juice promotes early insulin secretion [22], but green juice contains almost no carbohydrates or fats, and it is suggested that there is no mechanism mediated by insulin secretion. Greenleaf juice may be one of the most convenient and cost-effective methods for reducing postprandial blood glucose in healthy people.

## 5. Study Limitation

This study has some limitations that must be considered. First, we measured tissue glucose levels using a continuous blood glucose meter, not by laboratory analysis using venous blood sampling and did not measure insulin secretion. Second, blood glucose levels fluctuate with the sexual cycle in women; however, the sexual cycle of the participants was not adjusted during the examination. Third, the sample size was small, and sufficient statistical analysis was not possible.

## 6. Conclusions

We clarified the fluctuations in blood glucose levels if greenleaf juice was ingested simultaneously with rice and 30 min earlier. If greenleaf juice was ingested, postprandial blood glucose was suppressed at 90 or 120 min after meals, with no difference regardless of differences in the intake time. It has been suggested that drinking greenleaf juice may suppress postprandial blood glucose levels in healthy young individuals. Suppressing postprandial blood glucose levels may contribute to reducing the risk of developing vascular disorders.

## Figures and Tables

**Figure 1 nutrients-16-03226-f001:**
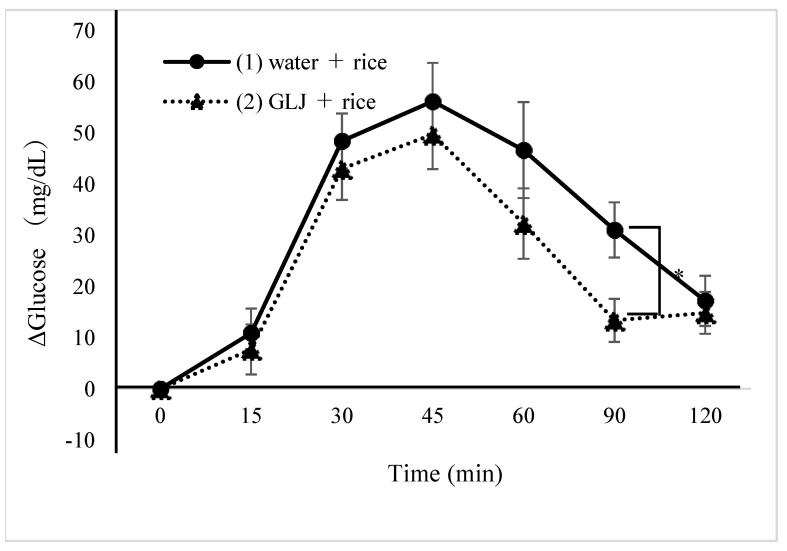
Δglucose value with simultaneous intake of greenleaf juice or water and rice. GLJ: green leaf juice, *: *p* < 0.05.

**Figure 2 nutrients-16-03226-f002:**
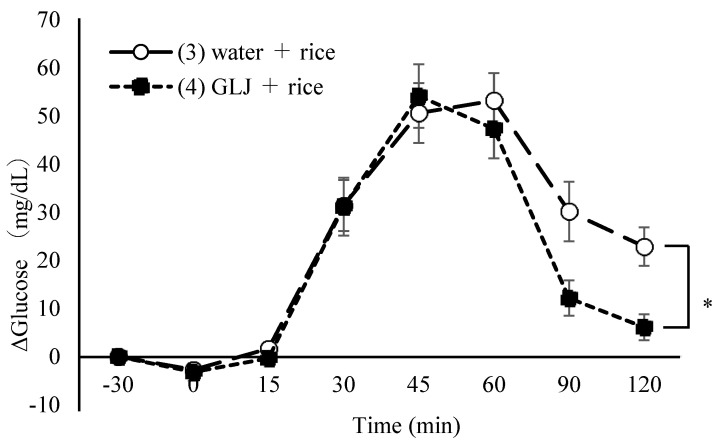
Δglucose value with intake 30 min before greenleaf juice or water and rice. *: *p* < 0.05, GLJ: green leaf juice.

**Table 1 nutrients-16-03226-t001:** Nutrient information for the greenleaf juice from this study.

Nutrition Information (per 200 mL)	
Energy (kcal)	0
Protein (g)	0
Fat (g)	0
Carbohydrate (g)	1.4
Sugars (g)	0
Dietary fiber (g)	0.3~1.4
Salt (g)	0~0.1
Potassium (mg)	16~131
Calcium (mg)	3~17
Iron (mg)	0.0~0.8
Vitamin E (mg)	8.4
Vitamin K (μg)	8~111
Folic acid (μg)	2~23
Caffeine (mg)	12

**Table 2 nutrients-16-03226-t002:** Test food amounts and nutritional components.

	Test Diet (1)·(3)	Test Diet (2)·(4)
	Water 200 mL + Rice 150 g	GLJ 200 mL + Rice 150 g
rice (g)	150	150
water (mL)	200	-
GLJ (mL)	-	200
Energy (kcal)	209	215
Sugars (g)	48.8	49.4
derived from rice (g)	48.8	48.8
derived from GLJ (g)	-	0.6
Lipid (g)	0	0
Protein (g)	2.9	2.9
Dietary fiber (g)	0.9	1.3–2.3
derived from rice (g)	0.9	0.9
derived from GLJ (g)	-	0.4–1.4

GLJ: greenleaf juice.

**Table 3 nutrients-16-03226-t003:** Changes in Δglucose levels after simultaneous ingestion.

Time (min)	(1) Water + Rice (mg/dL)	(2) GLJ + Rice (mg/dL)	*p*
0	0	0	-
15	10.9 ± 4.8	7.7 ± 4.9	N.S.
30	48.5 ± 5.4	43.1 ± 6.1	N.S.
45	56.2 ± 7.5	49.8 ± 6.8	N.S.
60	46.7 ± 9.4	32.3 ± 6.9	N.S.
90	31.1 ± 5.4	13.4 ± 4.2	0.030
120	17.2 ± 4.9	14.9 ± 4.1	N.S.

Values are shown as mean ± SEM (*n* = 13). GLJ: greenleaf juice. N.S.: not significant.

**Table 4 nutrients-16-03226-t004:** FPG, ΔBGmax, IAUC after simultaneous ingestion.

	(1) Water + Rice	(2) GLJ + Rice	*p*
FPG (mg/dL)	87.9 ± 3.0	88.3 ± 3.5	N.S.
ΔBGmax (mg/dL)	64.8 ± 7.5	56.5 ± 6.2	N.S.
IAUC	4009.6 ± 517.2	2946.4 ± 389.0	N.S.

Values are shown as mean ± SEM (*n* = 13). GLJ: greenleaf juice, FPG: fasting blood glucose levels, ΔBGmax: maximum postprandial blood glucose excursion, IAUC: incremental area under the curve. N.S.: not significant.

**Table 5 nutrients-16-03226-t005:** Changes in Δglucose levels after drinking before rice consumption.

Time (min)	(3) Water + Rice (mg/dL)	(4) GLJ + Rice (mg/dL)	*p*
−30	0	0	-
0	−2.8 ± 0.8	−3.2 ± 1.8	N.S.
15	1.6 ± 1.3	−0.4 ± 1.2	N.S.
30	31.4 ± 5.3	31.2 ± 6.0	N.S.
45	50.6 ± 6.2	54.2 ±6.6	N.S.
60	53.2 ± 5.7	47.3 ± 6.1	N.S.
90	30.2 ± 6.2	12.2 ± 3.7	N.S.
120	22.9 ± 4.0	6.1 ± 2.7	0.028

Values are shown as mean ± SEM (*n* = 13). GLJ: greenleaf juice. N.S.: not significant.

**Table 6 nutrients-16-03226-t006:** FPG, ΔGmax, IAUC after drinking before rice consumption.

	(3) Water + Rice	(4) GLJ + Rice	*p*
FPG (mg/dL)	87.8 ± 2.3	88.3 ± 2.5	N.S.
ΔBGmax (mg/dL)	59.9 ± 5.0	56.2 ± 6.0	N.S.
IAUC	2916.6 ± 320.9	2900.0 ± 339.9	N.S.

Values are shown as mean ± SEM (*n* = 13). GLJ: greenleaf juice, FPG: fasting blood glucose levels, ΔBGmax: maximum postprandial blood glucose excursion, IAUC: incremental area under the curve. N.S.: not significant.

## Data Availability

The datasets used and/or analyzed during this study are available from the corresponding author upon reasonable request.

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
