# Peer review of "Before or Concomitant Drinking Greenleaf Juice with Rice Reduces Postprandial Blood Glucose Levels in Healthy Young Women"

_nutrients, 2024, doi:10.3390/nu16193226_

Round 1
Reviewer 1 Report
Comments and Suggestions for Authors
There is a major concern in detecting food products that do not exert pressure on the pancreas, the level of postprandial glucose and implicitly do not produce an increase in insulin resistance. From my point of view, the proposed study has some weak points for which I do not see it at the level of the journal: the number of enrolled patients is very small, it was interesting to be designed in such a way that there is a control group, to present a more high age group (that is, a group of volunteers aged 45-65 years should be included because there the impact on postprandial blood sugar is better noted) and also a group of diabetic patients should be included, as well as male volunteers. In such conditions the results would have been much more conclusive, much more interesting comparisons could have been made and the value of the study would have been much greater. If the academic editor decides to accept this form of study, certain aspects must be developed:
- the introduction can be improved with much more detailed information related to the nutrients present in greenleaf juice (macro and micronutrients, possibly with concentrations and not just specifying them, it is insufficient to say vitamins and not indicate them, minerals, the type of fiber, the type of carbohydrates);
- in the material and methods section: a more detailed description of the study protocol, a description of the way in which the health status of the volunteers participating in the study was assessed, to specify why male subjects were not included;
- in the section where the limits of the study are specified, several things should be included: the lack of tests on elderly subjects, the lack of tests on male subjects, the small number of subjects enrolled in the study;
- in the conclusions section, specify the added value brought by this study.
Author Response
There is a major concern in detecting food products that do not exert pressure on the pancreas, the level of postprandial glucose and implicitly do not produce an increase in insulin resistance. From my point of view, the proposed study has some weak points for which I do not see it at the level of the journal: the number of enrolled patients is very small, it was interesting to be designed in such a way that there is a control group, to present a more high age group (that is, a group of volunteers aged 45-65 years should be included because there the impact on postprandial blood sugar is better noted) and also a group of diabetic patients should be included, as well as male volunteers. In such conditions the results would have been much more conclusive, much more interesting comparisons could have been made and the value of the study would have been much greater. If the academic editor decides to accept this form of study, certain aspects must be developed:
Thank you for your important comment.As you pointed out, I think it would be preferable to include middle-aged and elderly people and/or people with diabetes. This study was conducted on university students, so it was young women. This is also a pilot study, and we hope to expand the number of subjects in the future and measure blood glucose, insulin, incretins, etc. by blood collection, which could not be done this time.
the introduction can be improved with much more detailed information related to the nutrients present in greenleaf juice (macro and micronutrients, possibly with concentrations and not just specifying them, it is insufficient to say vitamins and not indicate them, minerals, the type of fiber, the type of carbohydrates);
In accordance with your suggestions, we have added the nutritional information table of the green juice used in the test as Table 1. (Page 2. 2-2 Test foods)
- in the material and methods section: a more detailed description of the study protocol, a description of the way in which the health status of the volunteers participating in the study was assessed, to specify why male subjects were not included;
All subjects were healthy young women. Previous studies have reported that there are gender differences in postprandial blood glucose variability in young adults with diabetes (Maria Gonzalez-Rodriguez et.al. Postprandial glycemic response in a non-diabetic adult population: the effect of nutrients is different between men and women. Nutrition &Metabolism (2019)16:46)- in the section where the limits of the study are specified, several things should be included: the lack of tests on elderly subjects, the lack of tests on male subjects, the small number of subjects enrolled in the study.
- in the conclusions section, specify the added value brought by this study.
In conclusion, it states that green juice may reduce the rise in blood sugar after meals. As an added value, it is stated that high postprandial blood glucose promotes arteriosclerosis in the background, so "Suppressing postprandial blood glucose elevation is expected not only to improve glycemic control in diabetics and reduce the onset and progression of diabetic complications, but also to reduce the risk of macrovascular complications in patients with no diabetes." I added.
Reviewer 2 Report
Comments and Suggestions for Authors
This great study shows the glycemia-lowering effect of greenleaf juice in people who have no diabetes (yet). The set-up and work-up are well sufficient for a pilot trial, even though no actual blood samples were taken. I have some comments:
- Was there no sign of insulin resistance in any of the patients?
- Acarbose has quite some gastro-intestinal side-effects, making it not really valuable to take in primary prevention.
Author Response
Comment to Reviewer 2
Comments and Suggestions for Authors
This great study shows the glycemia-lowering effect of greenleaf juice in people who have no diabetes (yet). The set-up and work-up are well sufficient for a pilot trial, even though no actual blood samples were taken. I have some comments:
- Was there no sign of insulin resistance in any of the patients?
As you pointed out, since fasting blood glucose and insulin levels have not been measured in blood samples, it is not possible to make a definitive decision. In order to eliminate insulin resistance associated with obesity as much as possible, those who fall under a BMI of 25 or higher, which is Japan's obesity standard, were excluded in the selection of subjects.
- Acarbose has quite some gastro-intestinal side-effects, making it not really valuable to take in primary prevention.
As you pointed out, acarbose has gastrointestinal side effects. Although it is approved as a drug covered by insurance in Japan, it is rarely used as a primary prevention. In animal experiments, it has been reported that the ingredients contained in green leaf have the α-glucosidase inhibitory effect as acarbose, so I think it makes sense to use such foods as primary prevention.
Reviewer 3 Report
Comments and Suggestions for Authors
This study examines the effects of Greenleaf juice on postprandial glycemia in young healthy women with BMI <25 kg/m2. Participants were glucose tolerant with normal postprandial glycemia. Selection of a sample with glucose intolerance would have complemented the introduction and discussion much better as most of these texts focus on postprandial hyperglycemia. These texts need to be modified to focus on the impact of reducing postprandial glycemia in a normoglycemic sample. In this study, the postprandial glycemia is only modestly impacted as might be expected with a glucose tolerant sample. The modest effects were noted using multiple paired t-tests which is not the proper statistical treatment for a repeated measures research design. A significant repeated measures ANOVA F test (examining the interaction between the 4 treatments) is required prior to post-hoc examination using t-tests.
Specific concerns:
1. The samples size (n=13) is not justified, particularly for a 4-arm study design. Please justify that there is adequate power to observe meaningful differences in a normoglycemic population.
2. Is this trial registered? Please provide that information.
3. Table 1: the headings appear to be inaccurate – should state: Test diet 1 and 3 and Test diet 2 and 3. Are the g of rice precooked or cooked. How was the rice prepared?
4. Why was a ~50 g carbohydrate load chosen? Typically for glycemia investigations 75 g or 100 g carbohydrate is used.
5. Treatment periods were only 2 days apart – typically postprandial glycemia trials use a 1-week separation. Furthermore, fasting is 12 hours, and participants are instructed to maintain their usual level of physical activity and diet throughout the study period and in particular the day prior to each test session. Were diet and exercise controlled the day prior to testing?
6. Were treatments randomly ordered? Was masking assured? These items need to be clearly discussed in methods.
7. The iAUC did not differ between groups. This suggests that the intervention did not reduce postprandial glycemia. It appears that the variability was large around the mean. The Figures need to include the variability measure (e.g., the SEM) to clearly depict the data variability.
8. In the abstract and conclusion, stating that these results can be expected in patients with type 2 diabetes is too speculative. As stated above, the text (introduction and discussion) need to focus on normoglycemic investigations and the relevance of these results for that population.
Comments on the Quality of English Languageminor editing of the English language is needed.
Author Response
Reviewer 3
Comments and Suggestions for Authors
This study examines the effects of Greenleaf juice on postprandial glycemia in young healthy women with BMI <25 kg/m2. Participants were glucose tolerant with normal postprandial glycemia. Selection of a sample with glucose intolerance would have complemented the introduction and discussion much better as most of these texts focus on postprandial hyperglycemia. These texts need to be modified to focus on the impact of reducing postprandial glycemia in a normoglycemic sample. In this study, the postprandial glycemia is only modestly impacted as might be expected with a glucose tolerant sample. The modest effects were noted using multiple paired t-tests which is not the proper statistical treatment for a repeated measures research design. A significant repeated measures ANOVA F test (examining the interaction between the 4 treatments) is required prior to post-hoc examination using t-tests.
Specific concerns:
Thank you very much for your many important comments.
As you pointed out, the subjects in this study were young non-diabetics, so we will add an effect on postprandial blood glucose levels in non-diabetic patients. According to the results of this study, the maximum postprandial blood glucose level peaked 30~45 minutes after eating, and some exceeded 140 mg/dL. Since some healthy young women present with postprandial hyperglycemia, it is important to correct postprandial hyperglycemia through diet.
As for statistics, ANOVA tests for Δglucose, ΔBGmax and IAUC were also performed. There was no significant difference between the four groups. However, in this study, the study design of test (1), (2) and test (3), (4) was different, and a statistical analysis between the two groups (t-test) was performed with water (1) or (3) as a control for postprandial blood glucose levels (Δglucose). We added the section on statistical analysis.
- The samples size (n=13) is not justified, particularly for a 4-arm study design. Please justify that there is adequate power to observe meaningful differences in a normoglycemic population.
As mentioned, postprandial blood glucose levels were assessed between two groups with the same study design. However, the sample size is not sufficient for either a two-group or four-group analysis. This study is a pilot study, and we believe that it is necessary to examine the number of subjects and the characteristics of the subjects in the future.
- Is this trial registered? Please provide that information.
This study is not registered in the database.
- Table 1: the headings appear to be inaccurate – should state: Test diet 1 and 3 and Test diet 2 and 3. Are the g of rice precooked or cooked. How was the rice prepared?
Thank you for pointing out the mistake in the heading of the table (changed to table 2 due to the addition of table 1). Fixed. The rice used in this study was cooked in packaged rice. It can be eaten as it is, but it was heated in the microwave just before intake.
- Why was a ~50 g carbohydrate load chosen? Typically for glycemia investigations 75 g or 100 g carbohydrate is used.
In Japan, 150 g of packaged rice (equivalent to 50 g of sugar) used as a basic diet in the glycemic index survey. In similar previous studies conducted in Japan, many of them used 150g or 200g of rice, and in this study, 150g of rice was used. This study focuses on how consuming green juice affects blood sugar in the normal diet of non-diabetics.
- Treatment periods were only 2 days apart – typically postprandial glycemia trials use a 1-week separation. Furthermore, fasting is 12 hours, and participants are instructed to maintain their usual level of physical activity and diet throughout the study period and in particular the day prior to each test session. Were diet and exercise controlled the day prior to testing?
In this study, fasting for more than 10 hours was considered fasting. Regarding the amount of activity on the day before the test, the subjects were informed to avoid exercise. I also instructed them to eat as much as they usually eat. In addition, the reason why the interval between the tests was short was that the sensor of the continuous blood glucose meter used was one that could measure for two weeks, and there was a concern that replacing the sensor in the middle of the test (the manufacturer indicated that the blood glucose results would differ slightly depending on the sensor), so we conducted four tests within two weeks.
- Were treatments randomly ordered? Was masking assured? These items need to be clearly discussed in methods.
The order of the trials was random, but masking could not be done due to the nature of the test diet.
- The iAUC did not differ between groups. This suggests that the intervention did not reduce postprandial glycemia. It appears that the variability was large around the mean. The Figures need to include the variability measure (e.g., the SEM) to clearly depict the data variability.
In accordance with your suggestions, I have included error bars in the figure.
- In the abstract and conclusion, stating that these results can be expected in patients with type 2 diabetes is too speculative. As stated above, the text (introduction and discussion) need to focus on normoglycemic investigations and the relevance of these results for that population
As you pointed out, it is necessary to conduct new tests on diabetics or people with impaired glucose tolerance to determine the effect on diabetics. In this study, I would like to limit myself to the expression that the effect of lowering postprandial blood glucose was inferred in non-diabetic young women without obesity.
Round 2
Reviewer 1 Report
Comments and Suggestions for Authors
I ask the authors to explain what karimu is, the element in the composition presented in table 1.
Author Response
Thank you for pointing this out. I'm sorry, this is a typo.
It is "karium" in Japanese, and it means “potassium”.
The amount of potassium is 16~131mg, and 0~0.1 is salt.
We corrected the table 1.
We also fixed a typo in the section at the bottom of the table. “caffein” →“Caffeine”
Reviewer 3 Report
Comments and Suggestions for Authors
Thank you for addressing my concerns. The statistical analyses are still concerning. As you are using multiple t-tests the Bonferroni correction for p values should be applied. Perhaps controlling for baseline value would be helpful? You would need to use univariate analyses to enter a control variable.
Author Response
Thank you for addressing my concerns. The statistical analyses are still concerning. As you are using multiple t-tests the Bonferroni correction for p values should be applied. Perhaps controlling for baseline value would be helpful? You would need to use univariate analyses to enter a control variable.
Thank you for your comment.
Regarding the multiplicity of tests that you pointed out, we have corrected the analysis of Tables 3 and 5 by making Bonferroni corrections. As a result, p-value of blood glucose after 90 minutes of tests (3) and (4) was no significant difference, and the Table 5 and the text were revised. The p-value after 90 minutes of (1) and (2) is 0.005 → 0.030, p-value after 120 minutes for groups (3) and (4) was 0.004 → 0.028, and Table 3.5 was revised.
Also, in the text Data Analysis section,、we added, “In the Δ-glucose analysis between the two groups, Bonferroni correction was performed for the obtained p-values in order to examine the multiplicity of tests.”
In addition, text Data Analysis section, we added, " In the Δ-glucose analysis between the two groups, Bonferroni correction was performed on the obtained p-values to examine the multiplicity of tests "